# Solid Sampling Pyrolysis Adsorption-Desorption Thermal Conductivity Method for Rapid and Simultaneous Detection of N and S in Seafood

**DOI:** 10.3390/molecules27248909

**Published:** 2022-12-15

**Authors:** Derong Shang, Wenyan Gu, Yuxiu Zhai, Jinsong Ning, Xuefei Mao, Xiaofeng Sheng, Yanfang Zhao, Haiyan Ding, Xuming Kang

**Affiliations:** 1Key Laboratory of Testing and Evaluation for Aquatic Product Safety and Quality, Ministry of Agriculture and Rural Affairs, Yellow Sea Fisheries Research Institute, Chinese Academy of Fishery Sciences, Qingdao 266071, China; 2Institute of Quality Standard and Testing Technology for Agro-Products, Chinese Academy of Agricultural Sciences, Key Laboratory of Agro-Food Safety and Quality, Ministry of Agriculture and Rural Affairs, Beijing 100081, China

**Keywords:** solid sampling, adsorption-desorption, thermal conductivity, seafood, N and S

## Abstract

In this work, a rapid method for the simultaneous determination of N and S in seafood was established based on a solid sampling absorption-desorption system coupled with a thermal conductivity detector. This setup mainly includes a solid sampling system, a gas line unit for controlling high-purity oxygen and helium, a combustion and reduction furnace, a purification column system for moisture, halogen, SO_2_, and CO_2_, and a thermal conductivity detector. After two stages of purging with 20 s of He sweeping (250 mL/min), N_2_ residue in the sample-containing chamber can be reduced to <0.01% to improve the device’s analytical sensitivity and precision. Herein, 100 s of heating at 900 °C was chosen as the optimized decomposition condition. After the generated SO_2_, H_2_O, and CO_2_ were absorbed by the adsorption column in turn, the purification process executed the vaporization of the N-containing analyte, and then N_2_ was detected by the thermal conductivity cell for the quantification of N. Subsequently, the adsorbed SO_2_ was released after heating the SO_2_ adsorption column and then transported to the thermal conductivity cell for the detection and quantification of S. After the instrumental optimization, the linear range was 2.0–100 mg and the correlation coefficient (R) was more than 0.999. The limit of detection (LOD) for N was 0.66 μg and the R was less than 4.0%, while the recovery rate ranged from 95.33 to 102.8%. At the same time, the LOD for S was 2.29 μg and the R was less than 6.0%, while the recovery rate ranged from 92.26 to 105.5%. The method was validated using certified reference materials (CRMs) and the measured N and S concentrations agreed with the certified values. The method indicated good accuracy and precision for the simultaneous detection of N and S in seafood samples. The total time of analysis was less than 6 min without the sample preparation process, fulfilling the fast detection of N and S in seafood. The establishment of this method filled the blank space in the area of the simultaneous and rapid determination of N and S in aquatic product solids. Thus, it provided technical support effectively to the requirements of risk assessment and detection in cases where supervision inspection was time-dependent.

## 1. Introduction

Nitrogen (Nitrogen, N) in aquatic products is an important source of high-quality proteins for humans [1,2,3,4], and the current detection method corresponding to the food protein content in China is the GB 5009.5-2016 National standard for food safety determination of protein in food [5]. This method needs to be decomposed by adding concentrated sulfuric acid at 420 °C under the condition of catalytic heating, and the ammonia produced combines with sulfuric acid to form ammonium sulfate. The ammonia is separated by alkalization distillation, and titrated with a sulfuric acid or hydrochloric acid standard titration solution after absorption with boric acid. The nitrogen content is calculated according to the consumption of acid, and then multiplied by the conversion factor to obtain the content of protein. The international detection methods of N content include the thermal conductivity method, the Kjeldahl Method, the Dumas combustion method, the Sei-micro Kjeldahl Method [6,7,8,9,10,11], and so on. The Kjeldahl method, with its simple operation and relatively low equipment cost, is still recognized by almost all countries and organizations today [6]. However, lots of chemical reagents, such as concentrated sulfuric acid, sodium hydroxide solution, and boric acid solution are required, which not only increases the cost of testing, but also contaminates the environment seriously and can cause potential harm to the health of humans [12,13,14]. The Dumas combustion method has the advantages of small sample consumption, with no sample digestion process and no reagent pollution, all while being faster and more efficient [14,15]; however, the popularity of the Dumas combustion method is far smaller than that of the Kjeldahl method for the moment, due to the high cost of the instruments [15,16].

In order to ensure the health of consumers, China has established specific regulations for food additives, including sulfur dioxide. It limits the scope of the SO_2_ used in food and the maximum allowable residue dose. The desire to make food brighter leads to the an exceedance of sulfur dioxide in food products, which affects the health of consumers [17,18,19]. The current detection method corresponding to SO_2_ in China is the GB 5009.34-2016 National standard for food safety determination of sulfur dioxide in food [20]. This standard is only applicable to dry food. In this method, the sample is acidified and distilled in a closed container, and the distillate is absorbed with a lead acetate solution. The absorbed solution is acidified with hydrochloric acid, and then titrated by the iodine standard solution. The sulfur dioxide content in the sample is calculated according to the amount of iodine standard solution consumed. The Infrared Spectroscopy method, high-temperature combustion method, and acid–base titration method, Eschka Method, and Coulomb Titration Method were often used to detect the amount of S content [21,22,23,24,25]. When S was detected by the infrared method, different S infrared pools were needed for high S content and low S content. The high-temperature combustion method, acid–base titration method, and Eschka Method have the advantages of high accuracy and good repeatability, with disadvantages such as time consumption and cumbersome operation. The Infrared Spectroscopy method and Coulomb Titration Method are used widely with the advantages of high accuracy, convenient operation, high automaticity, and short time [26].

The above detection methods were for the determination of either N or S, including different detectors and different instruments. Meanwhile, most detection methods required cumbersome pretreatment steps such as digestion or distillation and titration, resulting in many problems such as long detection time and environmental pollution during digestion or distillation. To the best of our knowledge, there is no standard method or relevant instrument for an adsorption-desorption thermal conductivity method for the simultaneous detection of N and S in seafood in China thus far. There are similar instruments abroad, but there is no method standard. The United States, Australia, Germany, Britain, and other countries have conducted in-depth research on coal elemental analysis methods [27,28,29,30,31,32]. Therefore, the research and establishment of a rapid detection method for solid injection are in full swing [33,34,35,36,37,38,39,40,41,42].

The solid sampling analytical method of N and S in seafood was established based on the principle of adsorption-desorption thermal conductivity analysis, in which only one thermal conductivity detector was employed for the instrumental integration. In addition, this setup is mainly composed of a solid sampling system, a gas line unit for controlling high-purity oxygen and helium, a combustion and reduction furnace, and a purification column system. After optimization, the limits of detection (LODs) for N and S were 0.66 μg and 2.29 μg, equivalent to 6.6 μg/g and 22.9 μg/g at a 100 mg sample size, respectively. The total time of analysis was less than 6 min without the sample preparation process, fulfilling the fast detection of N and S in seafood. 

## 2. Results and Discussion

### 2.1. The Design of Solid Sampling System

This proposed setup mainly includes a solid sampling system, a gas line unit for controlling high-purity oxygen and helium, a combustion and reduction furnace, a purification column system for moisture, halogen, SO_2_, and CO_2_, and a thermal conductivity detector. One single thermal conductivity cell was employed to measure N and S simultaneously, unlike the previous methods that only measure nitrogen or sulfur [5,6,7,8,9,10,11,12,13,14,15,16,20,21,22,23,24,25,26,27,28,29,30,31,32,33,34,35,36,37,38]. To fulfill this goal, two-stage purging was newly designed and the relatively high-temperature decomposition and carrier gases were all optimized herein.

#### 2.1.1. Design of Two-Stage Purging

In the previous studies, one-stage purging always resulted in excessive nitrogen residue in the sample falling hole, thereby impacting the following measurement of S. So, we designed a two-stage purging to fulfill the simultaneous detection of N and S. Herein, the first stage of purging occurs when the sample enters the sample falling hole of the ball cup; then, the ball cup was rotated to make the sample falling hole face the inlet pipe, isolating it from the outside air. The first stage of purging can reduce the N_2_ residue in the container to 1% at the highest. Subsequently, the second stage of purging was carried out to remove the residual N. After two stages of purging, N_2_ residue in the sample-containing chamber can be reduced to <0.01% so as to avoid excessive nitrogen residue. As with the previous studies, helium, a kind of inert gas, was chosen as the purging gas with an appropriate flow rate to ensure that the wrapped sample would not be deformed or damaged. As a result, compared to the previous one-stage purging methods, the two-stage purge design gives the lowest N signal as shown in Figure 1, while effectively eliminating the air introduction into the sample falling hole. 

#### 2.1.2. High-Temperature Decomposition

Real seafood samples must be ashed and decomposed to release analytes containing N and S for subsequent measurement. To investigate the effect of decomposition, a nori CRM (GBW10023) with N content (5.0 ± 0.3%) and S content (2.26 ± 0.14%) was employed under different decomposition temperatures and times, and the results are shown in Figure 2. The measured contents of N and S go up with the increase in decomposition temperature from 600 to 800 °C; then, a plateau appears from 800 to 1000 °C, indicating a complete release of analytes containing N and S from the sample matrix. The curves of the decomposition time for N and S both demonstrate the highest signals at 100 s heating at 900 °C, thereby revealing a sufficient sample decomposition, measured by observing the sample residue after heating. Considering the favorable analysis precision (<5.0%), 100 s heating at 900 °C was chosen as the optimized decomposition condition. 

#### 2.1.3. Carrier Gas

The carrier gas flow rate has an effect on the adsorption capacity of the adsorption column. The higher the flow rate, the lower the adsorption capacity. When the flow rate is too low, the sampling time becomes longer. The carrier gas pressure has an impact on the adsorption ability of the adsorption column. The greater the pressure, the stronger the adsorption ability. A high level of pressure results in high requirements for the furnace bearing force and pipeline sealing. If the pressure is too low, the components cannot be completely adsorbed and the adsorption efficiency is reduced, which affects the results.

Based on the content of the tested sample nori CRM (GBW10023), the N content (5.0 ± 0.3%) and S content (2.26 ± 0.14%) were used to compare the response value changes between different Carrier gas flow rates (n = 6). The results are shown in Figure 3. A lower content, including 0.4% N and 2.2% S, was determined at a gas flow rate of 250 mL/min when compared to the standard value. A lower content, 21.6% and 51.3%, and 17.0% and 22.1% at gas flow rates of 200 mL/min and 300 mL/min for N and S, respectively, showed the worst accuracy. As such, a gas flow rate of 250 mL/min was chosen as the best.

### 2.2. Establishment of Standard Curves

Standard substance nori CRM (GBW10023) was weighed at 2.24 mg, 4.25 mg, 5.71 mg, 10.81 mg, 16.51 mg, 21.78 mg, 25.40 mg, and 31.61 mg and then decomposed in tinfoil cups placed in a Ni-Cr chrome electric resistance furnace. The generated H_2_O and CO_2_ were adsorbed by two purification columns. NOx was reduced to N_2_ in a Ni-Cr chrome electric resistance furnace and N_2_ was measured by a high-purity helium-loaded thermal conductivity detector.

The standard curve was generated by plotting the N and S concentrations versus the fluorescent area of the electric resistance furnace. Calculated from the absolute content versus the signal intensities (peak area) of N and S, the equation obtained for N was: y = 0.0000643396x − 0.0187, R = 0.9991. This standard curve is shown in Appendix A. The equation obtained for S was: y = 0.000197x − 0.0606, R = 0.9998, shown in Appendix A. Appendix A represents the standard curve of N and S.

### 2.3. Linearity Range, Recovery Rate, Precision, and Sensitivity of the Method

The linear range was 2.0–100 mg and the correlation coefficient (R) was more than 0.999. The limit of detection (LOD) for N was 0.66 μg, equivalent to 6.6 μg/g at a 100 mg sample size (maximum sample size for sample boat), calculated from 3σ of 11 measurements close to blank. The R was less than 4.0% and the recovery rate ranged from 95.33% to 102.8%. At the same time, the LOD for S was 2.29 μg, equivalent to 22.9 μg/g at a 100 mg sample size (maximum sample size for sample boat), calculated from 3σ of 11 measurements close to blank. The R was less than 6.0% and the recovery rate ranged from 92.26% to 105.5%. The total time of analysis was less than 6 min.

The proposed method met the requirements of accurate and rapid N and S detection by the direct injection of aquatic product solids. The analytical performance of this instrument was in full accordance with the N and S content requirements of the assay, as shown in Table 2 and Table 3. In Table 4 and Table 5, the recovery rates of the N and S content of Pectinidae (GBW10024) (n = 12.8 ± 0.8%; s = 1.5 ± 0.1%) (n = 6) ranged from 100.2 to 102.8% and 96.66 to 103.3% and RSD was less than 5.0%. For *Fenneropenaeus chinensis* (GBW10050) (N = 13.5%; S = 1.0%) and ground fish powder, better assay accuracy was shown in Appendix A, respectively. The recovery rate for N in *Fenneropenaeus chinensis* (GBW10050) and ground fish powder ranged from 95.34 to 101.3% and 97.31 to 100.2%, respectively. The recovery rate for S ranged from 92.26 to 100.2% and 96.93 to 105.5%, respectively.

### 2.4. Detection and Validation of Real Sample

The Spirulina (GBW10025) (n = 10.6 ± 0.4%; s = 0.78 ± 0.08%) and shredded squid bought from malls were used to compare the detection results obtained in this study with the current method of testing N (GB 5009.5-2016) and S (GB 5009.34-2016) in food products in Table 6. The ANOVA test showed no significant difference (*p* > 0.05). In Appendix A, more than 50 real samples including the types of dry, fresh, and aquatic products were detected to verify the methodological ability. The ANOVA test showed no significant difference (*p* > 0.05). It showed that this method was better than the national standard method due to the instruments and instrument condition parameters that we researched, and the selection was very reasonable and appropriate.

## 3. Experimental Materials and Methods

### 3.1. Instrumentation

This solid sampling nitrogen and sulfur analyzer (5e-chons2400) (Figure 4) mainly comprises a solid sampling system, a gas line unit for controlling high-purity oxygen and helium, a two-stage purge device, a combustion and reduction furnace, a purification column system for moisture, halogen, SO_2_, and CO_2_, and a thermal conductivity detector. The thermal conductivity cell section was equipped with a mass flow controller (MFC), gas path control system, and a reference gas flow controller and detector. The samples enter the high-temperature decomposition catalytic module for oxidative decomposition with oxygen, producing CO_2_, NO_X_, H_2_O, SO_X_, halide, and other gases. Then, in the reduction module, NO_X_ and SO_X_ react with agent copper particles to generate N_2_ and SO_2_, respectively. Excess O_2_ is absorbed through the reaction with copper particles to form CuO. The halogen compound is purified by the silver wire in the reduction tube. So far, the gas coming from the reduction tube only contains N_2_, CO_2_, H_2_O, and SO_2_. The mixed gas in the instrument is separated and purified successively by the adsorption-separation method (Figure 4 and Figure 5). The mixed gas passes through SO_2_, H_2_O, and CO_2_ adsorption devices to be absorbed in turn, and N_2_ is left, and then N_2_ is detected by the thermal conductivity cell for the quantification of N. Subsequently, SO_2_ can be obtained by heating the SO_2_ adsorption device after switching to the air path of SO_2_, and then SO_2_ is detected by the thermal conductivity cell for the quantification of S.

### 3.2. Chemicals and Standards

Samples were purchased from the National Standard Center of China (Beijing, China), which included nori CRM (GBW10023, certified N values = 5.0 ± 0.3%, certified S values = 2.26 ± 0.14%), Pectinidae (GBW10024, certified N values = 12.8 ± 0.8%, certified S values = 1.5 ± 0.1%), Spirulina (GBW10025, certified N values = 10.6 ± 0.4%, certified S values = 0.78 ± 0.08%), and Fenneropenaeus chinensis (GBW10050, certified N values = 13.5%, certified S values = 1.0%). Furthermore, other dry or freshwater products were acquired from local supermarkets (Qingdao, China). 

### 3.3. Analytical Procedures by Solid Sampling System

The analytical procedures of the solid sampling system are summarized in Table 1, including two-stage purging, high-temperature decomposition, adsorption-desorption, and detection. (1) Two-stage purging: the sample was weighed into the sampling boat and after 20 s of He sweeping (250 mL/min) to reduce nitrogen residue. (2) High-temperature decomposition: the sample was inserted into the high-temperature decomposition catalytic module at 900 °C pyrolysis for 100 s for oxidative decomposition with oxygen, producing CO_2_, NO_X_, H_2_O, SO_X_, halide, and other gases. (3) Adsorption-desorption: The mixed gas was transported into the bottom of the combustion tube with reduced copper to generate N_2_ and SO_2_ and excess O_2_ was absorbed. Subsequently, the mixed gas passes through SO_2_, H_2_O, and CO_2_ adsorption devices to be absorbed in turn. SO_2_ was adsorbed by the adsorption column fitted with carbon molecular sieves; H_2_O was captured by the first gas purification column filled with magnesium perchlorate and phosphorus pentoxide; CO_2_ was captured by the second purification column packed with alkali asbestos and phosphorus pentoxide. Finally, N_2_ was left to be detected. (4) N Detection: N_2_ was detected within 6 s by the thermal conductivity cell for the quantification of N under the conditions of 8 KΩ thermal resistance, a 7× amplification circuit, 250 mL/min carrier gas, and 50 mL/min reference gas. (5) S Detection: After completing N Detection, the captured SO_2_ was released once the SO_2_ adsorption column was heated, and then it was transported to the thermal conductivity cell for the detection and quantification of S. 

### 3.4. Sample Preparation

The edible sites from living fresh aquatic products were made into homogenate using a T18 homogenizer (IKA, Königswinter, Germany). The dry aquatic product was pulverized to a powder for use. The fresh sample, with a weight of about 50.00 mg, was weighed using a tinfoil cup using a Cp2250 electronic analytical balance (sartorius, Goettingen, Germany), and the powder sample weighed about 10 mg (depending on its N and S content). Then, their N and S content was determined in the machine.

## 4. Conclusions

In this work, a novel solid sampling method for the rapid and simultaneous detection of N and S in seafood was for the first time developed. The two-stage purging method was newly designed to avoid excessive nitrogen residue, so as to improve analytical sensitivity and precision. Thus, the method LOD for N and S reached 0.66 μg and 2.29 μg, respectively. The recovery rate of N and S ranged from 95.33 to 102.8% and 92.26 to 105.5%, respectively. The whole analysis time was <6 min without the sample preparation process. Herein, only one thermal conductivity detector was employed for further instrumental integration. This method has promising application potential in the sensitive, rapid, green, and robust detection of N and S to evaluate the nutritional quality of seafood. 

## Figures and Tables

**Figure 1 molecules-27-08909-f001:**
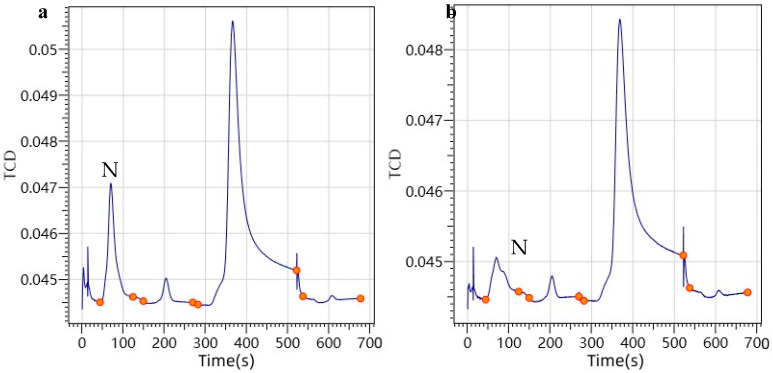
Blank comparison diagram of one-stage purge and two-stage purge. For Panel (**a**), under the condition of one-stage purging, the N signal is high, and the N brought from the air is not completely purged. For Panel (**b**), under the condition of two-stage purge, we can see the lowest and most stable N signal.

**Figure 2 molecules-27-08909-f002:**
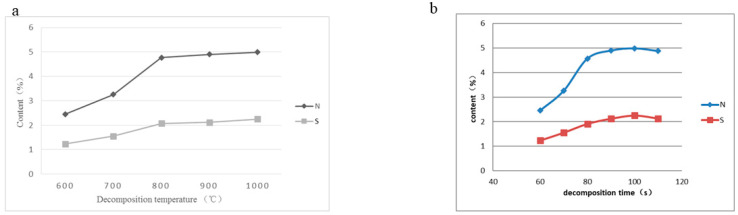
Effect of decomposition temperature and time on N and S contents. The GBW10023 CRMs of nori had a N concentration of 5.0 ± 0.3% and a S concentration of 2.26 ± 0.14%. For Panel (**a**), it was measured by the TCD using different decomposition temperatures. For Panel (**b**), it was measured using different decomposition times. Other experimental conditions were performed as per Table 1.

**Figure 3 molecules-27-08909-f003:**
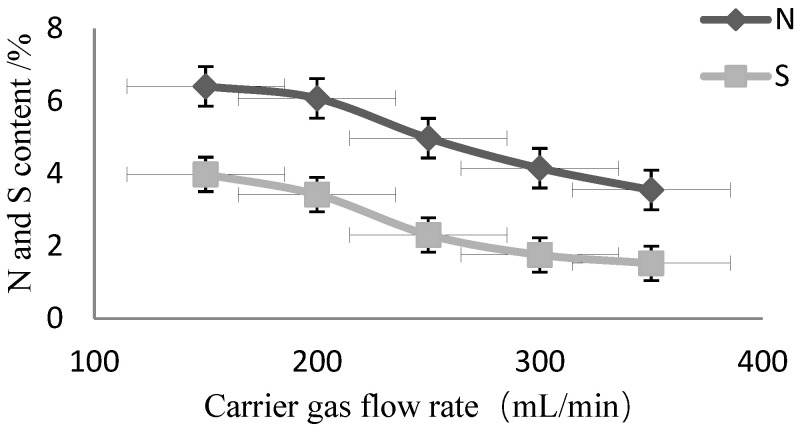
Effect of Carrier gas flow rate on N and S content. (n = 6) Herein, the CRM of nori (GBW10023) was measured by the TCD using different gas flow rates. Other experimental conditions were performed as per Table 1.

**Figure 4 molecules-27-08909-f004:**
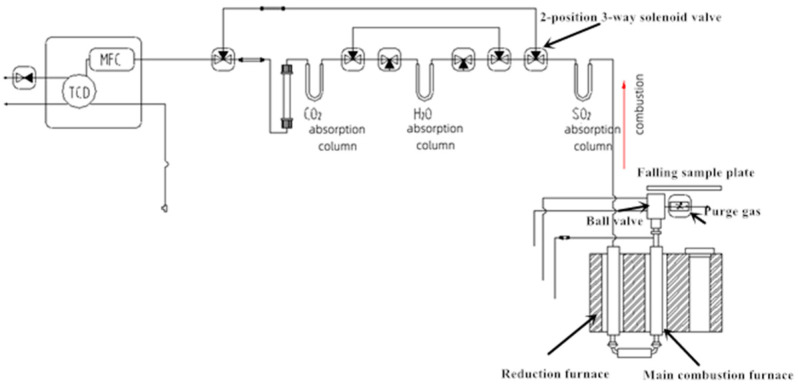
Schematic diagram of solid sampling nitrogen and sulfur analyzer. TCD means a thermal conductivity detector; MFC means mass flow controller.

**Figure 5 molecules-27-08909-f005:**
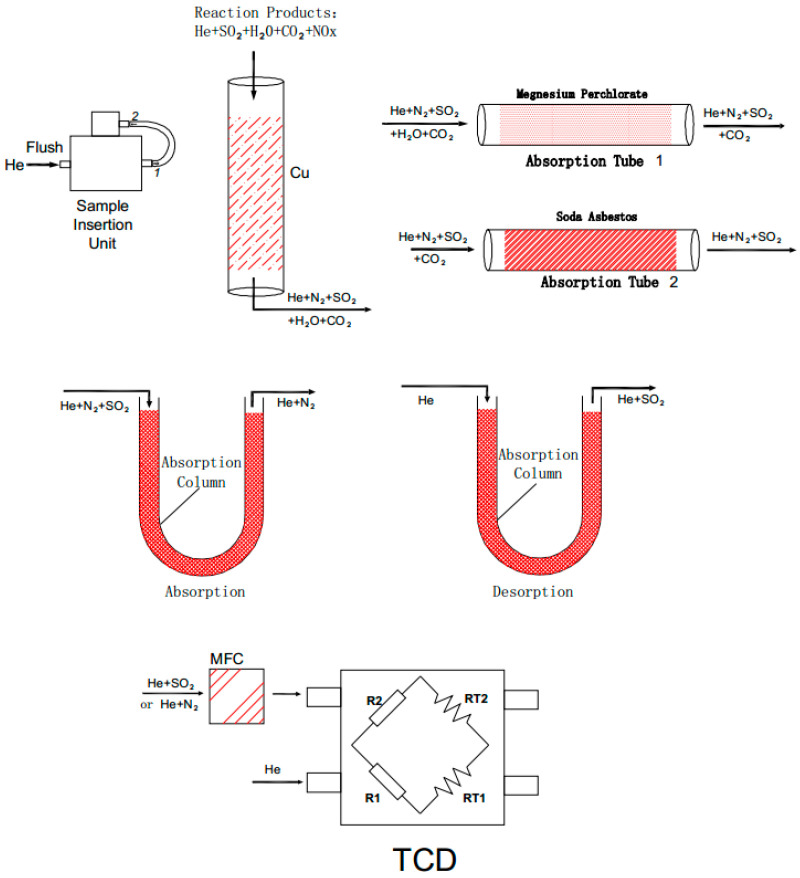
Schematic diagram of adsorption-desorption gas path.

**Table 1 molecules-27-08909-t001:** Instrumental program of solid sampling system.

Solid Sampling Analyzer			TCD	
Program	Time (s)	Temperature (°C)		
The first purge	20	/	Thermal resistance	8 KΩ
The second purge		/	Amplification circuit	7×
High-temperature decomposition	100	900	Carrier gas flow rate (mL/min)	250
Adsorption tube 1	Mg(ClO4)_2_, P_2_O_5_ (absorb H_2_O)		Reference gas flow rate (mL/min)	50
Adsorption tube 2	Soda asbestos, P_2_O_5_ (absorb CO_2_)		Reading time (s)	6
Adsorption column	Carbon molecular sieves (absorb SO_2_)			

**Table 2 molecules-27-08909-t002:** N content analytical performance of the instrument.

Linear Range (mg)	R	LOD (μg)	RSD (%)
2.0–100	>0.9999	0.66	3.79

**Table 3 molecules-27-08909-t003:** S content analytical performance of the instrument.

Linear Range (mg)	R	LOD (μg)	RSD (%)
2.0–100	>0.9998	2.29	5.44

**Table 4 molecules-27-08909-t004:** Results of N recovery rate (%) in Chlamys farre standard substances (n = 6).

*Pectinidae* CRM (GBW10024)	Substance Number	Detection Result (g/100 g)	Standard Reference Values (g/100 g)	Recovery (%)
	GB10024		12.8 ± 0.8	
1	ditto	12.86	ditto	100.5
2	ditto	12.83	ditto	100.2
3	ditto	12.89	ditto	100.7
4	ditto	13.16	ditto	102.8
5	ditto	13.11	ditto	102.4
6	ditto	12.95	ditto	101.1

**Table 5 molecules-27-08909-t005:** Results of S recovery rate (%) in Chlamys farre standard substances (n = 6).

*Pectinidae* CRM (GBW10024)	Substance Number	Detection Result (g/100 g)	Standard Reference Values (g/100 g)	Recovery (%)
	GB10024		1.5 ± 0.1	
1	ditto	1.49	ditto	99.33
2	ditto	1.55	ditto	103.3
3	ditto	1.5	ditto	100
4	ditto	1.47	ditto	98
5	ditto	1.45	ditto	96.66
6	ditto	1.49	ditto	99.33

**Table 6 molecules-27-08909-t006:** Comparison between this method and national standard methods for N and S content detection in food (n =6).

Detection Method	Weighed Sample (mg)	Time of Digestion or Distillation (h)	Detection Time (h)	GBW10025 (N = 10.6 ± 0.4%; S = 0.78 ± 0.08%)	RSD (%)	Shredded Squid (%)	RSD (%)	Recovery (%)
This method	15.00	0	0.1	N = 10.05	0.35	N = 4.98	1.28	94.0–101.8
				S = 0.76	5.68	S = 0.41	3.67	91.0–103.8
GB 5009.5-2016 (N)	600.0	2.5	4.5	N = 10.02	0.28	N = 4.97	0.40	94.2–99.1
GB 5009.34-2016 (S)	5000	1.0	2.5	S = 0.79	5.38	S = 0.40	7.80	94.9–107.6

## Data Availability

The data presented in this study are available on request from the corresponding author. The data are not publicly available due to the privacy of the study participants.

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
