# Peer review of "Solid Sampling Pyrolysis Adsorption-Desorption Thermal Conductivity Method for Rapid and Simultaneous Detection of N and S in Seafood"

_molecules, 2022, doi:10.3390/molecules27248909_

Round 1
Reviewer 1 Report
This manuscript developed a quick approach for the simultaneous detection of N and S in seafood on the basis of a solid sampling absorption-desorption device coupled with a thermal conductivity detector. The manuscript is well organized and written. Also, the study objective is worth for publication. This reviewer strongly encourages authors to address the concerned issues of this manuscript. Thus, the manuscript need minor amendments before publication. Constrictive comments for revision are:
Minor comments:
Line 22: Please use comma (,) after H2O
Line 31: Please use ‘and’ instead of &
Line 100: Please use the suitable and meaningful word after EXPERIMENTAL.
Line 109: Please use comma (,) after halide
Line 115: Please use comma (,) after H2O
Line 132: You should use similar format when you incorporate unit (either space or not space) throughout the manuscript.
Line 135: Please use comma (,) after halide
Line 138: Please use comma (,) after H2O
Line 231: Fenneropenaeus chinensis- It should be italic
Author Response
Dear Reviewers:
Thank you so much for valuable comments. We have made modification carefully with your advices on the original manuscript, with which we hope to meet the publication standards.
Please see the attachment.

Reviewer 2 Report
Review report
Title: It looks very general. So I would suggest to put an innovative title which can be needed with question marks. So I would recommit its reorientation.
Abstract: Line no 33-34 should be rewritten and it’s not a recommendation of study. Abstract should provide a flow of objective, material methods, results and significant recommendation. So it should be revised thoroughly accordingly.
Introduction: The references should be replaced with latest references preferably should be beyond 2017 at least 80 percentage references should be like that.
Material and methods
There is need to mention the procedure whether they are as per international standard or modified from exiting one.
Results
There is need to comply the results with material and methods and with discussions of the study so it will be better to replicate it in future by other researchers.
Conclusion is not impressive so it should be clear cut finding which can provides the outcome of the study.
Overall good attempt but there is high need of making coherence between material methods , results and discussion so that it will be better to follow up by other researchers. In many places sentence error, typographical mistakes in symbols etc. In addition a through revision is need for improving the languages es many sentence needs rephrasing to make it easily understandable.
Author Response

(The authors gave the same response as above.)

Reviewer 3 Report
The submitted article ‘Rapid and simultaneous detection of N and S in seafood using solid sampling pyrolysis adsorption-desorption thermal conductivity method’ is interesting and good article. This is a kind of rare studies, hence is very desire. It will be appropriate for MOLECULES (MDPI). Below I pointed most of mistakes and matters for explanation.
1. In introduction the strong emphasis should be placed on justification
2. The good idea should be summary as graphical workflow in introduction because this topic is complex, perhaps chart/graph will be good idea
3. Conclusion part should be more informative (more details, perhaps in brackets?)
4. Please include advances and disadvantages of your studies in conclusions
In general this manuscript is well prepared. Perhaps from technical point of view it will be better to change order, i.e.
MATERIALS AND METHODS AFTER RESULTS AND DISCUSSION. Graphs and tables should be included in the text after text, not after the references.
However, the quality of data are appropriate
I totally agree that this is very important subject, and it is very important for publication in MDPI. I will recommend this article for publication in MOLECULES after minor revision.
Author Response

(The authors gave the same response as above.)
